# Multi-Task Learning via Scale Aware Feature Pyramid Networks and Effective Joint Head

## Abstract

As a concise and classic framework for object detection and instance segmentation, Mask R-CNN achieves promising performance in both two tasks. However, considering stronger feature representation for Mask R-CNN fashion framework, there is room for improvement from two aspects. On the one hand, performing multi-task prediction needs more credible feature extraction and multi-scale features integration to handle objects with varied scales. In this paper, we address this problem by using a novel neck module called SA-FPN (Scale Aware Feature Pyramid Networks). With the enhanced feature representations, our model can accurately detect and segment the objects of multiple scales. On the other hand, in Mask R-CNN framework, isolation between parallel detection branch and instance segmentation branch exists, causing the gap between training and testing processes. To narrow this gap, we propose a unified head module named EJ-Head (Effective Joint Head) to combine two branches into one head, not only realizing the interaction between two tasks, but also enhancing the effectiveness of multi-task learning. Comprehensive experiments show that our proposed methods bring noticeable gains for object detection and instance segmentation. In particular, our model outperforms the original Mask R-CNN by 1~2 percent AP in both object detection and instance segmentation task on MS-COCO benchmark. [1]

## 1 Introduction

In the past few years, object detection and instance segmentation results were rapidly improved by the powerful baseline system Mask R-CNN(He et al. (2017)), which extends Faster R-CNN(Girshick (2015)) by adding a branch for predicting an object mask in parallel with the existing part for bounding box recognition. This method is conceptually natural and offers extensibility and robustness, shows a surprisingly smooth, flexible, and fast system for instance segmentation results.

However, this remarkable multi-task learning method suffers from a common problem of modern detection methods. That is scale variation, since Convolutional Neural Network is sensitive to scales. And what's more, performing multi-tasks needs more credible feature extraction execution and multi-scale complementary features integration. Therefore, it is urgent to tackle this problem. Feature pyramid is a common practice. FPN(Lin et al. (2017a)) augmented a top-down path with lateral connections for object detection. It exploits the inherent multi-scale, pyramidal hierarchy of deep convolutional networks to construct feature pyramids with marginal extra cost. Rethinking the extracted multi-scale features of general FPN, the top-down pathway FPN only introduces high-level semantic information to low-level feature, while ignore the role of low-level feature for localization. FPN still has room for improvement.

Another aspect for improvement of Mask R-CNN in multi-task learning is about the parallel isolated branches of Mask R-CNN. The segmentation branch of Mask R-CNN is based on the output of Region Proposal Network (RPN) in training stage, which ignores the inherently tie in those two tasks and is inconsistent with testing processes. In common sense, instance segmentation is connected detection based on the bounding box strictly, which is more meticulous than the bounding box. However, the bounding box is easy to obtain than the masking label. It is worthy of trying to explore and enhance the interrelations between object detection and instance segmentation.

---

[1]Code will be available soon.

In this paper, we make a natural extension of Mask R-CNN architecture, merging the detection branch and the instance segmentation branch into single branch. This smart framework contribution of our work named EJ-Head (Effective Joint Head), including three operations: "Interleaving","Enriched Feature" and "Boundary Refinement". EJ-Head promotes both two tasks consistently and provides an example for improving multi-task learning.

In general, we improve two aspects in Mask R-CNN by proposing SA-FPN (Scale Aware Feature Pyramid Networks) and EJ-Head (Effective Joint Head). Experimental results on the challenging COCO benchmark show that when using our proposed modules, detection and instance segmentation performances are improved by about 1.6 and 1.4 percent AP increment in box AP and mask AP, respectively.

The main contributions of our work highlighted as follows. (1) We propose SA-FPN (Scale Aware Feature Pyramid Networks), which effectively integrates multi-scale complementary feature and solves the problem of scale variation in an innovative way. (2) We slickly mix the mask branch and detection branch into one branch and introduce EJ-Head (Effective Joint Head), which can reinforce each task and also eliminate the gap between training and testing processes. (3) We propose a newly enhanced Mask R-CNN, which provides a reference and is helpful for further research on multi-task learning.

## 2 RELATED WORK

### 2.1 DEEP OBJECT DETECTORS

Deep learning based methods(Girshick et al. (2014), He et al. (2015b)) have tremendously pushed forward the remarkable progress in object detection over a short period of time. Mainstream object detection frameworks roughly fall into two categories. Two-stage methods like Faster R-CNN(Girshick (2015)), R-FCN(Dai et al. (2016b)), Mask R-CNN(He et al. (2017)) generate a sparse set of candidate proposals that contain all objects while filtering out the majority of negative locations in the first stage, and then classify the proposals into foreground classes or background in the second stage. Single-stage approaches, such as SSD(Liu et al. (2016)),YOLO(Redmon & Farhadi (2016)),RetinaNet(Lin et al. (2017b)) directly regress to predict the bounding boxes. Detection frameworks with multiple stages like Cascade R-CNN(Dai et al. (2016a)) are also popular and bring tremendous improvement for object detection.

### 2.2 SCALE VARIATION

Scale variation across object instances has been treated as one of the most knotty problem in modern development of detection. To address this challenge, several approaches have been proposed. Image pyramid is an intuitive way, SNIP (Singh & Davis (2018)) and SNIPER(Singh et al. (2018)) select a specific scale for each resolution during multi-scale training. However, these methods suffer a lot from the inevitable increase of inference time. Instead of taking multiple images as input, feature pyramid method uses multi-level features of different layers. FPN(Lin et al. (2017a)) is the most famous representative of this strategy. It takes the fused feature map with the highest resolution to pool features and achieves superior performance. SSD(Liu et al. (2016)), DSSD(Fu et al. (2017)) and MS-CNN(Cai et al. (2016)) perform object detection at multiple layers for objects of different scales. PANet(Liu et al. (2018)) boosts information flow in proposal-base instance segmentation framework, enhances the entire feature hierarchy with accurate localization signals in lower layers by bottom-up path augmentation, which shortens the information path between lower and top layers feature.

### 2.3 INSTANCE SEGMENTATION

Instance segmentation is a task to predict class label and pixel-wise instance mask in an image. There are mainly two streams of methods in instance segmentation, segmentation based methods and detection based methods. The former are less commonly used nowadays. Segmentation based methods like (Zhang et al. (2016)) first predict the labels of each pixel and then identify object instances therefrom. However the performances of these methods are always unsatisfactory.

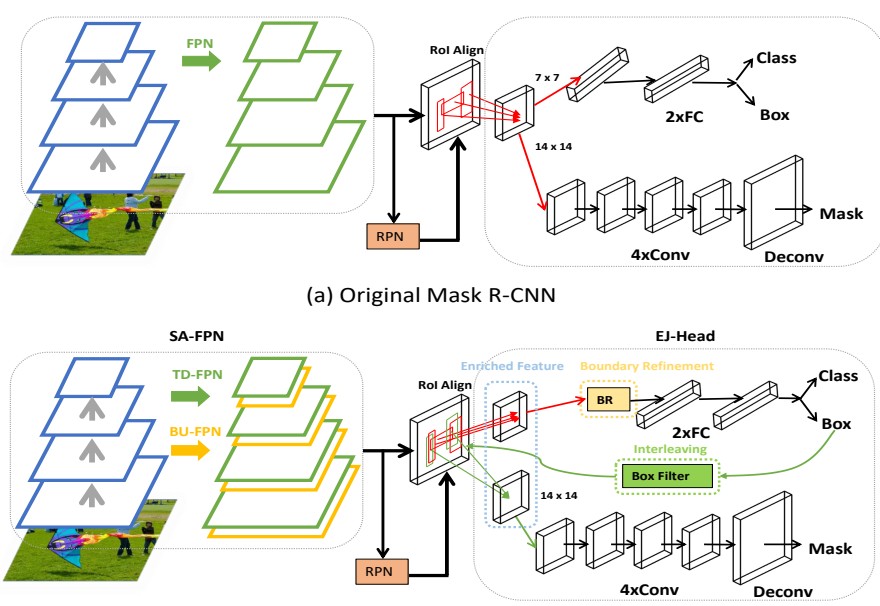

Figure 1: Architectures of the original Mask R-CNN in (a) and our proposed model in (b). Obviously, our model is different from the original model by reformimg two modules. SA-FPN (Scale Aware Feature Pyramid Networks) combines TD-FPN (Top-Down style FPN, as shown in green) and BU-FPN (Bottom-Up style FPN, as shown in yellow) together. And EJ-Head (Effective Joint Head) proposes three operations. "Enriched feature" represents enhencing the extracted RoI feature, "Boundary Refinement" means adding additional convolutions on this pathway for optimization of the boundary , "Interleaving" is an operation to filter the predicted bounding boxes which get a high IoU (Intersection-over-Union) with ground truth and then feed positive samples into the instance segmentation branch.

Detection based methods follow a similar diagram: get the region of each instance and then predict the mask, showing a strong connection to object detection. Instance-FCN(Dai et al. (2016a)) proposes instance-sensitive FCNs to generate the position-sensitive to obtain the final masks. FCIS(Li et al. (2017)) takes position-sensitive maps with inside/outside scores to generate the instance segmentation results, but exhibits systematic errors on overlapping instances and creates spurious edges. MaskLab(Chen et al. (2018b)) produces instance-aware masks by combining semantic and direction predictions. Cascade Mask R-CNN is a multi-stage object detection and instance segmentation framework derived from Cascade R-CNN (Cai & Vasconcelos (2018)), which comprises multiple stages where the output of each stage is fed into the next one for higher quality refinement. But this multi-stage mechanism inevitably brings extra computation overhead of inference.

## 3 FRAMEWORK

In this section, we will describe our proposed new framework for object detection and instance segmentation in details.

**Overview.** Based on Mask R-CNN, as shown in Figure 1, it is distinctive in two aspects: (1) It aims to remedy the problem of scale variation by fusing Top-Down style FPN and Bottom-Up style FPN into a novel neck module called SA-FPN. (2) It explores the interrelations between object detection and instance segmentation, eliminating the gap between training and testing processes and enhancing both two tasks.

Overall, these changes to the framework architecture effectively improve two tasks and also provides a reference for further research on multi-task learning.

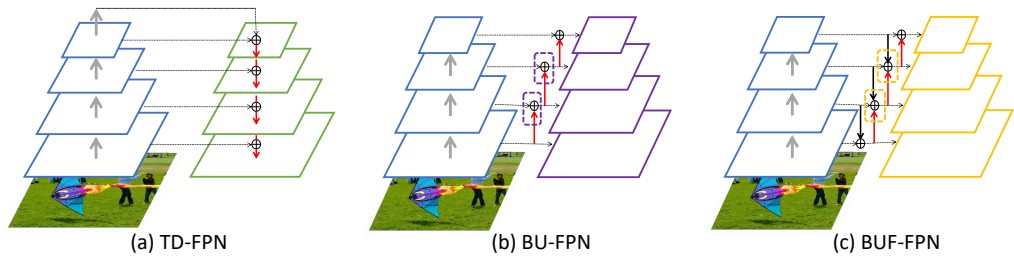

Figure 2: Three different architectures of FPNs. (a) is the most widely used Top-Down style FPN, which is used in the baseline model. (b) and (c) are two newly designed Bottom-Up style FPNs. The main difference is that in (b) each layer of feature pyramid combines with only neighboring shallower layer, while in (c) both neighboring shallower and deeper features are fused. So we call (c) Bottom-Up-Fusion FPN (BUF-FPN). Details are shown in purple dotted boxes in (b) and in orange dotted boxes in (c). Experiments in Table 3 show that BU-F FPN performs better. And when we concatenate (a) and (c) together and add 1x1 convolutions to reduce dimensions, we propose SA-FPN (Scale Aware FPN), as illustrated in Figure 1 (a).

## 3.1 SCALE AWARE FEATURE PYRAMID NETWORKS

Scale variation across object instances is one of the most challenging problems in object detection, especially for very small or huge objects. Simply using multi-scale image pyramids has some improvement in accuracy but suffers a lot from increasing inference time. Feature pyramid is popular to deal with this knotty problem, and Top-Down style FPN in (Lin et al. (2017a)) is the most widely used feature pyramid structure.

As illustrated in Figure 2 (a), a general FPN is a Top-Down style structure that feeds the last output of the backbone into the top of feature pyramid, and the information flow direction is from shallow to deep in the backbone and from the top to bottom in feature pyramid. Up-sampling operation and lateral connection fusion are applied step by step to get a larger feature until the bottom of the pyramid. Thus, deep feature map with high semantic information and shallow feature map with high resolution are combined. This greatly boosts the performance on small objects detection and therefore improves the overall detection performance.

However, the top-down pathway of general FPN only introduces high-level semantic information to low-level feature, but ignore the subsidiary role of low-level feature like edges and textures which also are important for accurately localizing instances. Most of the improvements caused by Top-Down FPN come from more accurate detection on small objects. Shallow features with high resolution contain useful low-level features. Different layers are of different resolutions, the fusion operation between neighbor shallower or deeper feature is also important.

Following the discussion above, we design two new Bottom-Up style FPNs, as illustrated in Figure 2 (b) BU-FPN and (c) BUF-FPN. Experiments in Table 3 show that (c) BUF-FPN performs better. The main difference between (b) and (c) is the feature integration operation, Details are shown in purple dotted boxes in (b) and in orange dotted boxes in (c). In (b) each layer of feature pyramid combines with only neighboring shallower layer, while in (c) both neighboring shallower and deeper layers. So we use (c) BUF-FPN as the default Bottom-Up style FPN for further experiments.

BU-FPN takes every level output of the backbone as the input of the next step and applies down-sampling and fusion operation step by step to build a feature pyramid from bottom to top. Letting low-level information guide high-level semantic information, BU-FPN is especially better at accurate detection of large objects.

When we combine TD-FPN (Top-Down FPN) and BUF-FPN (Bottom-Up-Fusion FPN) together, a scale sensitive neck module called SA-FPN (Scale Aware FPN) is proposed. Experiments in Table 3 show that concatenating performs better than element-wise addition. Then we add 1x1 convolutions to reduce dimensions. In line with expectations, SA-FPN absorbs the advantages of two style FPNs and improves both detection and instance segmentation in different object scales.

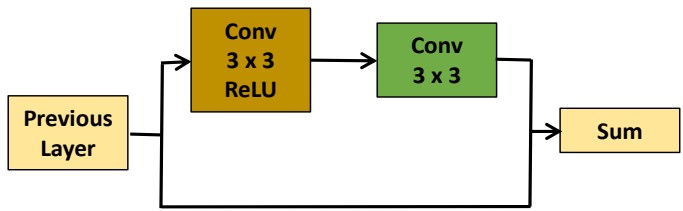

Figure 3: Detail of Boundary Refinement(BR) module in EJ-Head. There are two branches, one is two stacked convolutions, the other is a shortcut. Finally two branches are added at the pixel level, which brings the boundary alignment effect.

## 3.2 EFFECTIVE JOINT HEAD.

For multi-task learning, parallel execution of tasks is a common practice. However, instance segmentation in Mask R-CNN is such a detection based method that relies heavily on detection. Thus, the interaction between these two tasks is worth further exploring.

Firstly, we simply connect the segmentation branch behind the detection head in series when training, feeding the bounding box predictions of the detection into the mask head. As shown in Figure 1 (b), green dotted boxes "Interleaving" represents this execution. "Interleaving" is an operation to filter predicted bounding box which have IoU(Intersection over Union) with ground truth boxes of at least 0.5 and feed positive samples of mask into instance segmentation branch. In this way, the segmentation branch can take advantage of the detection process, realizing the interaction between two tasks. Moreover, the training and testing pipelines are highly consistent. And experiments in Table 4 prove this operation indeed improved performance.

Merging heavy mask branch into box branch may slightly increase the burden of box branch. We enlarge the extracted RoI features from 7 x 7 to 14 x 14, and name it "Enriched Feature" as shown in blue dotted boxes in Figure 1 (b). The quality of RoI features is enhanced by the enlarged receptive field. Richer features further help enhance both the detection and instance segmentation. Experiments in Table 4 show the effectiveness of our operation.

The whole flow of information goes through an unified path. Despite increased resolution, we can add additional convolutions on this pathway to further boost localization especially for details and edges of objects. As shown in Figure 3 the yellow dotted boxes "Boundary Refinement" means this execution. Boundary Refinement module is a residual structure for boundary alignment. Experiments in Table 5 show that "Boundary Refinement" is more superior than directly two stacked 3 x 3 simple convolutions together.

## 4 EXPERIMENTS

### 4.1 DATASETS AND EVALUATION METRICS

**Datasets Description.** We perform all experiments on MS-COCO(Lin et al. (2014)), which is the most representative and challenging dataset for object detection as well as instance segmentation. Following COCO-2017 settings, we train our models on 2017*train* (115k images) and present experimental results on 2017*val* and 2017*test-dev*.

**Evaluation Metrics.** We report the standard COCO-style evaluation metric AP (averaged over IoU thresholds) on two tasks, including $AP_{50}$, $AP_{75}$ (AP at different IoU thresholds) and $AP_S$, $AP_M$, $AP_L$ (AP at different scales). Both box AP and mask AP are evaluated.

### 4.2 IMPLEMENTATION DETAILS

In all experiments, we use ResNet(He et al. (2015a)) as backbones. In experiments based on ResNet-50, we use 8 NVIDIA TITAN Xp GPUs (2 images per GPU). To ensure the consistency of the overall batch size for heavier backbone network ResNet-101, we utilize 16 GPUs (one image per GPU) in

our experiments. The input shape of images are resized to 1333 and 800 for training and testing. We train all models with PyTorch(Paszke et al. (2017)) and mmdetection(Chen et al. (2018a)). We adopt different training strategies to save the training time. Specifically, unless noted, for ablation studies, we train for 12 epochs with an initial learning rate of 0.02, and decrease it by 0.1 after 9 and 11 epochs. While for the models used to compare with state-of-the-art methods, we train for 20 epochs with the same learning rate, and decrease it by 0.1 after 16 and 19 epochs.

### 4.3 BENCHMARKING RESULTS

We compare our model with the state-of-the-art object detection and instance segmentation approaches on the COCO dataset. In Table 1, we can see that our method exhibits substantial improvements compared to the single task (*i.e.*, object detection or instance segmentation) methods. In particular, our model is 4.5% absolutely better than Faster R-CNN(Ren et al. (2015)) (with RoI Align) in the criterion of box AP. Our method also outperforms FCIS(Li et al. (2017)) model by 3.0% in mask AP.

Table 1: Comparison with state-of-the-art methods on COCO *test-dev* dataset. Note that Our main research is about single-stage methods. Note that "Res101" means ResNet-101.

| Method | Backbone | box AP | $AP_{50}$ | $AP_{75}$ | mask AP | $AP_{50}$ | $AP_{75}$ |
|---|---|---|---|---|---|---|---|
| Faster R-CNN | Res101-FPN | 38.0 | 58.7 | 40.6 | - | - | - |
| FCIS | Res101 | - | - | - | 29.2 | 49.5 | - |
| FCIS+ | Res101 | - | - | - | 33.6 | 54.5 | - |
| MaskLab | Res101 | 39.6 | 60.2 | 43.3 | 35.4 | 57.4 | 37.4 |
| MaskLab+ | Res101 | 41.9 | 62.6 | 46.0 | 37.3 | 59.8 | 39.6 |
| Mask R-CNN | Res101-FPN | 40.9 | 62.3 | 44.3 | 37.0 | 59.1 | 39.4 |
| PANet | Res101-FPN | **42.8** | 64.0 | **46.4** | 38.0 | 60.5 | 40.4 |
| **Ours** | Res101-FPN | 42.5 | **64.0** | 46.1 | **38.4** | **60.6** | **40.9** |

We then conduct the comparisons with multi-task learning methods that jointly predict object bounding boxes and segmentation masks. Notably, comparing with Mask R-CNN, our model reports 1.6% and 1.4% gains in terms of box AP and mask AP, respectively. Obviously, our model performs better than MaskLab+(Chen et al. (2018b)). Compared with PANet(Liu et al. (2018)) which is the state-of-the-art single-stage based method, our model performs even a little better. Although the proposed model performs slightly worse than Cascaded Mask R-CNN, the multi-stage method consumes more computing resources and time.

### 4.4 ABLATION STUDY

**Component-wise Analysis.** To evaluate the effectiveness and generalization ability of two main components (*i.e.*, SA-FPN and or EJ-Head), we conduct comparative experiments on different backbones and methods. From Table 2, we can see that the SA-FPN module improves the box AP and mask AP by 0.6% and 0.5% respectively, compared to the Mask R-CNN methods based on ResNet-50-FPN backbone. Accordingly, the EJ-Head contributes to 1.0% and 0.7% improvement under the same settings. The combination of SA-FPN and EJ-Head modules brings significant improvements by 1.7% and 1.4% respectively, demonstrating that these two modules work in a complementary manner. Besides, consistent improvement is achieved based on the ResNet-101 backbone. It is also worthy to note that the proposed SA-FPN and EJ-Head modules also provide considerable improvements for multi-stage method like Cascaded Mask R-CNN, showing that our method can serve as plugin units for existing methods.

**SA-FPN Design.** In this paper, SA-FPN module is proposed to address the problem of scale variation. As shown in Table 3, TD-FPN performs better for small objects while BU-FPN and BUF-FPN are better for large objects. BUF-FPN fuses neighboring feature of both higher and lower level performs better than BU-FPN which only exploits neighboring feature from lower layer. And concatenating performs better then element-wise addition, so we add 1x1 convolutions to reduce dimensions. This demonstrating the bi-directional fusion manner obtains better feature representation in Bottom-Up style FPN. Our SA-FPN module conbines Top-Down style FPN (TD-FPN) and Buttom-

Table 2: Effects of each component in our design. Results are reported on COCO 2017*val*. Note that "Res50" means ResNet-50, "Res101" means ResNet-101.

| Model | Backbone | box AP | mask AP |
|---|---|---|---|
| Mask R-CNN (Baseline) | Res50-FPN | 37.2 | 34.1 |
| Mask R-CNN + SA-FPN | Res50-FPN | 37.8 | 34.6 |
| Mask R-CNN + EJ-Head | Res50-FPN | 38.2 | 34.8 |
| **Mask R-CNN + SA-FPN + EJ-Head (Ours)** | Res50-FPN | **38.9** | **35.5** |
| Mask R-CNN | Res101-FPN | 39.4 | 35.9 |
| **Mask R-CNN + SA-FPN + EJ-Head (Ours)** | Res101-FPN | **41.0** | **37.3** |
| Cascade Mask R-CNN | Res50-FPN | 41.3 | 35.7 |
| Cascade Mask R-CNN + SA-FPN + EJ-Head | Res50-FPN | **42.8** | **36.9** |

Up style FPN (BUF-FPN) and obtains $0.7\%$ and $0.6\%$ improvements respectively, indicating that the combination facilitates the handling of objects with varied scales.

Table 3: Ablation study of FPN designs on COCO 2017*val*. BU-FPN (shown in Figure 2 (b)) means using the fusion operation in Bottom-Up style FPN to compensate shallow feature by fusing only shallower neighboring feature. While BUF-FPN (shown in Figure 2 (c)) fuse both higher and lower level neighboring feature. $AP_S$, $AP_M$, $AP_L$ means AP at small, middle and large scales. Note that in the following discussion, we treat BUF-FPN as the default Buttom-Up style FPN.

| FPN design | box AP | $AP_S$ | $AP_M$ | $AP_L$ | mask AP | $AP_S$ | $AP_M$ | $AP_L$ |
|---|---|---|---|---|---|---|---|---|
| TD-FPN | 37.2 | 33.8 | 56.7 | 65.0 | 34.1 | 28.0 | 52.4 | 62.8 |
| BU-FPN | 34.6 | 30.0 | 53.0 | 65.2 | 31.8 | 24.8 | 49.0 | 63.0 |
| BUF-FPN | 36.8 | 32.0 | 55.9 | **66.1** | 33.7 | 26.6 | 50.9 | **63.9** |
| TD-FPN add BUF-FPN | 37.0 | 32.8 | 56.3 | 65.2 | 33.9 | 27.1 | 51.3 | 63.0 |
| **SA-FPN** | **37.9** | **34.5** | **57.4** | 65.6 | **34.8** | **29.4** | **52.7** | 63.3 |

**EJ-Head Design.** Next we investigate the effect of key ingredients of EJ-Head including "Enriched feature", "Boundary Refinement" and "Interleaving". As shown in Table 4, each component performs better than the Mask R-CNN. In particular, "Boundary Refinement" module brings the most significant improvement over baseline model by $0.6\%$ for box AP, indicating that the quality of RoI is enhanced by the enlarged receptive field. In addition, "Interleaving" with "Enriched Feature" exhibits the superior improvements over $0.6\%$ for mask AP, which verifies the effectiveness of the proposed modules. Overall, EJ-Head achieves $1.1\%$ and $0.7\%$ on box AP and mask AP, respectively.

In Table 5 we study the various designs of "Boundary Refinement" module in EJ-Head. The proposed Boundary Refinement module performs better than directly stacked convolutions. The residual structure plays an important role for boundary alignment. Stacked convolutions also have a little bit boost, but not very obvious as Boundary Refinement. When stacked more convolutions, it even will harm the performance of mask prediction.

Table 4: Ablation study of EJ-Head on COCO 2017*val*. The original Mask R-CNN uses none of three operations as shown in the first row in the table. Note that "Int" means "Interleaving", "EF" means "Enriched Feature", "BR" means "Boundary Refinement".

| Method | Int | EF | BR | box AP | mask AP |
|---|---|---|---|---|---|
| Mask R-CNN(Baseline) | | | | 37.2 | 34.1 |
| Mask R-CNN | ✓ | | | 37.2 | 34.5 |
| Mask R-CNN | | ✓ | | 37.6 | 34.4 |
| Mask R-CNN | | | ✓ | 37.8 | 34.2 |
| Mask R-CNN | ✓ | ✓ | | 37.7 | 34.7 |
| Mask R-CNN + EJ-Head | ✓ | ✓ | ✓ | **38.3** | **34.8** |

Qualitative results of multi-task learning results are illustrated in Figure 4. These results are based on ResNet-101-FPN, achieving a box AP of 42.5 and mask AP of 38.4. Masks are shown in color, bounding boxes and categories are also shown. It can be clearly seen that people and cars of different

Table 5: Ablation study of the design of "Boundary Refinement" on COCO-2017*val*. "Simple Conv" means using common 3 x 3 stacked convolutions. "2 x Simple Conv + Shortcut" is shown in Figure 3.

| Boundary Refinement (BR) | box AP | mask AP |
|---|---|---|
| None | 37.2 | 34.1 |
| 1 x Simple Conv | 37.4 | 34.1 |
| 2 x Simple Conv | 37.6 | 34.1 |
| 3 x Simple Conv | 37.6 | 34.0 |
| 2 x Simple Conv + Shortcut | **37.8** | **34.2** |

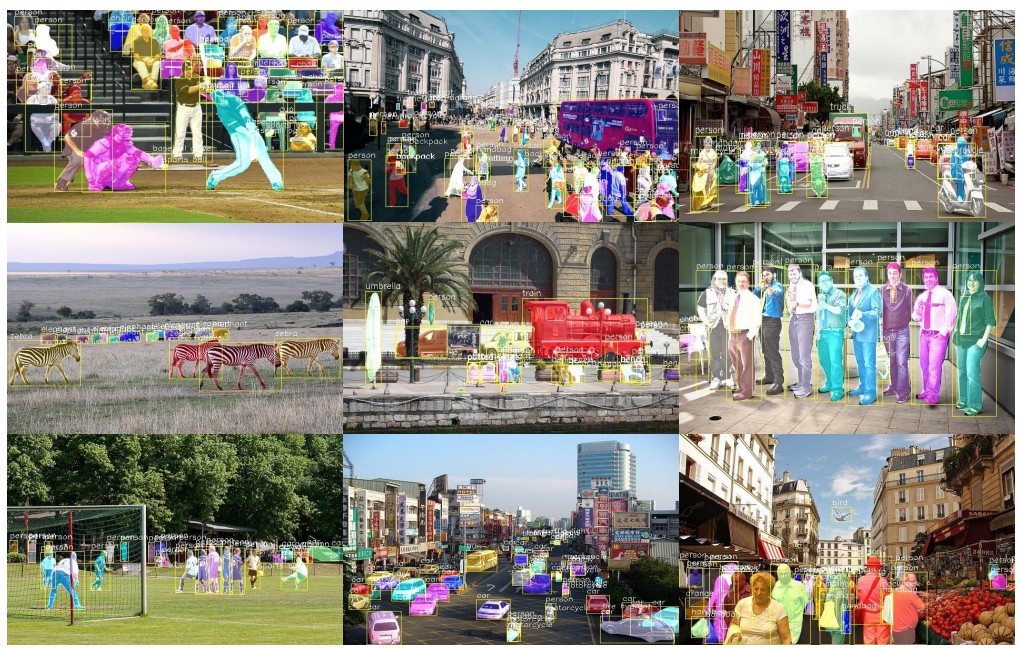

Figure 4: Examples of multi-task learning results on COCO 2017*test-dev*. Predicted detection results are shown in yellow bounding boxes, masks are also shown.

scales achieve accurate detection and robust instance segmentation, which shows the effectiveness of our methods. Moreover, in real-world scenarios such as autonomous driving, video surveillance and even in the wild, small objects are well parsed and sensed, and large objects have clear contour boundaries and detailed information.

## 5 CONCLUSIONS AND FUTURE WORK

In this paper we improve Mask R-CNN by proposing SA-FPN and EJ-Head for multi-task learning. This framework progressively integrates complementary multi-scale features in SA-FPN and enhances both detection and instance segmentation tasks with EJ-Head. It shows an innovative way to remedy scale variation issue, and also interweaves detection and segmentation branches for multi-task learning. Without bells and whistles, our overall system obtains remarkable improvements on COCO test-set, achieving 42.5 box AP and 38.4 mask AP. We hope our simple and effective approach will serve as a new baseline and contribute to both object detection and instance segmentation.

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
