# OpenReview forum: "Multi-Task Learning via Scale Aware Feature Pyramid Networks and Effective Joint Head"
_ICLR.cc/2020/Conference — Reject_

### Official Review · AnonReviewer3 · 2019-10-23
**Official Blind Review #3**

**Rating:** 3

**Review:**

This paper works on the problem of improving object detection and instance segmentation. It is realized by two independent contributions: 1). adding a high-to-low-resolution connection in FPN and 2) adding more connecting layers between the mask and classification heads. Experiments show both contributions give a small improvement (~1AP) on both detection and instance segmentation task.

Overall, the method seems reasonable and the improvements are healthy. The main concern is the technical novelty. It is not supervising that adding more connections inside the network can improve some performance. People have tried a lot of that (e.g. M2Det). These kinds of improvements come with a cost of slowdown and are usually not that appealing in practice.

The motivation for leveraging the relationship between segmentation and detection as a joint model is interesting and relatively new. However, the proposed method of feeding back the regressed bbox for segmentation seems straightforward and far from the full potential. It also requires a larger RoI feature map, which makes the contribution less clear. A fancier method or a more thorough analysis of how the information is shared between the two tasks is demanded for an ICLR publication.

**Experience Assessment:**

I have published one or two papers in this area.

**Review Assessment: Checking Correctness Of Derivations And Theory:**

I did not assess the derivations or theory.

**Review Assessment: Checking Correctness Of Experiments:**

I did not assess the experiments.

**Review Assessment: Thoroughness In Paper Reading:**

I read the paper thoroughly.

---

### Official Review · AnonReviewer2 · 2019-10-24
**Official Blind Review #2**

**Rating:** 3

**Review:**

The paper proposes several modifications to the Mask R-CNN model:
1. a variation of Feature Pyramid Networks (FPN): SA-FPN which merges features top-down and bottom-up.
2. the "effective joint head" (EJ-Head) which consists of
    a) moving the segmentation head behind the detection one,
    b) Doubling resolution of RoI crops, calling this "enriched feature".
    c) adding Boundary Refinement (really just a residual block).
Experiments show slightly improved scores on MS-COCO


I vote to reject this paper.

First of all, it is badly written. Not only are grandiose formulations of "smart framework", "innovative way", "we slickly mix" not professional, but the whole write-up of the method is very confusing. After reading it multiple times, I am still not 100% sure I fully understood the SA-FPN.

Second, it is not clear to me that SA-FPN is really novel, the original FPN paper already compared top-down and bottom-up performances (Tables 1-3).

Then, two modifications which simply increase capacity and could explain all improved scores are not ablated: increased resolution of RoI crops, and added "boundary refinement", which is really just a residual block.

Furthermore, moving the segmentation *after* the bounding-box prediction is *not* joint prediction. If predicting the box is called p(b) and predicting the mask is called p(m), the Original Mask R-CNN does p(b|features)p(m|features), the proposed model in this paper does p(b|features)p(m|b,features), and actual joint prediction would be p(b,m|features).

Finally, I think the paper is much better suited for a conference like ICCV/ECCV or CVPR and will get better reviewers than me there.

**Experience Assessment:**

I do not know much about this area.

**Review Assessment: Checking Correctness Of Derivations And Theory:**

N/A

**Review Assessment: Checking Correctness Of Experiments:**

I assessed the sensibility of the experiments.

**Review Assessment: Thoroughness In Paper Reading:**

I read the paper at least twice and used my best judgement in assessing the paper.

---

> ### Comment · AnonReviewer2 · 2019-11-15
> **Mistake in my final comment of the review**
>
> I apologize for a mistake in the penultimate paragraph of my above review. Of course, p(b|features)p(m|b,features) = p(m, b | features) and thus it *is* joint prediction.
>
> I am sorry for this very basic mistake!

---

### Decision · Program_Chairs · 2019-12-19

**Decision:**

Reject

**Comment:**

All three reviewers gave scores of Weak Reject. Only a brief rebuttal was offered, which did not change the scores. Thus the paper connect be accepted.